

# Fatigue and perceptual responses of heavier- and lighter-load isolated lumbar extension resistance exercise in males and females

Charlotte Stuart[1], James Steele[1], Paulo Gentil[2], Jürgen Giessing[3] and James P. Fisher[1]

[1] School of Sport, Health and Social Sciences, Southampton Solent University, United Kingdom
[2] Faculty of Physical Education and Dance, Universidade Federal de Goiás, Brazil
[3] Institute of Sport Science, University of Koblenz-Landau, Germany

## ABSTRACT

**Background**. There is a lack of research considering acute fatigue responses to high- and low-load resistance training as well as the comparison between male and female responses. Furthermore, limited studies have considered fatigue response testing with the inclusion of perceptions of discomfort and exertion.

**Methods**. The present study included males ($n = 9$; $23.8 \pm 6.4$ years; $176.7 \pm 6.2$ cm; $73.9 \pm 9.3$ kg) and females ($n = 8$; $21.3 \pm 0.9$ years; $170.5 \pm 6.1$ cm; $65.5 \pm 10.8$ kg) who were assessed for differences in fatigue (i.e., loss of torque at maximal voluntary contraction (MVC)) immediately following isolated lumbar extension (ILEX) exercise at heavy- (HL) and light-(LL) loads (80% and 50% MVC, respectively). Participants also reported perceptual measures of effort (RPE-E) and discomfort (RPE-D) between different resistance training protocols.

**Results**. Analysis of variance revealed significantly greater absolute and relative fatigue following LL compared to HL conditions ($p < 0.001$). Absolute fatigue significantly differed between males and females ($p = 0.012$), though relative fatigue was not significantly different ($p = 0.160$). However, effect sizes for absolute fatigue (HL; Males $= -1.84$, Females $= -0.83$; LL; Males $= -3.11$, Females $= -2.39$) and relative fatigue (HL; Males $= -2.17$, Females $= -0.76$; LL; Males $= -3.36$, Females $= -3.08$) were larger for males in both HL and LL conditions. RPE-E was maximal for all participants in both conditions, but RPE-D was significantly higher in LL compared to HL ($p < 0.001$) with no difference between males and females.

**Discussion**. Our data suggests that females do not incur the same degree of fatigue as males following similar exercise protocols, and indeed that females might be able to sustain longer exercise duration at the same relative loads. As such females should manipulate training variables accordingly, perhaps performing greater repetitions at a relative load, or using heavier relative loads than males. Furthermore, since lighter load exercise is often prescribed in rehabilitation settings (particularly for the lumbar extensors) it seems prudent to know that this might not be necessary to strengthen musculature and indeed might be contraindicated to avoid the increased fatigue and discomfort associated with LL exercise.

Corresponding author
James P. Fisher,
james.fisher@solent.ac.uk

## INTRODUCTION

Strengthening of the lumbar extensors using isolated lumbar extension (ILEX) resistance exercise is evidenced to be achievable using a low-volume (single-set) and low-frequency (1 day/week) approach in both asymptomatic trained males (*Fisher, Bruce-Low & Smith, 2013*; *Steele et al., 2015*) and males and females symptomatic of chronic low back pain (CLBP; *Bruce-Low et al., 2012*; *Steele, Bruce-Low & Smith, 2015*). A growing body of research supports that deconditioning of the lumbar extensor musculature (e.g., decreased strength and greater fatigability) are factors associated with CLBP (*Steele, Bruce-Low & Smith, 2014*). As such, strengthening of the lumbar extensors, where isolation is achieved using a restraint system preventing rotation of the pelvis, has been recommended to condition (i.e., strengthen, improve fatigability) this muscle group (*Steele, Bruce-Low & Smith, 2013*; *Gentil, Fisher & Steele, 2017*), and has been shown to reduce CLBP (*Bruce-Low et al., 2012*; *Steele, Bruce-Low & Smith, 2015*).

A definition of fatigue presented, and generally accepted, in the literature is "*an acute impairment of performance that includes both an increase in the perceived effort necessary to exert a desired force and the eventual inability to produce this force*" (*Enoka & Stuart, 1992*; page 1631). From this it has been hypothesised that the rate of muscle fatigue should coincide with the force requirements of the specific task to equate to a 100% value (*Morton, McGlory & Phillips, 2015*). For example, if exercise were performed at ~70% MVC to momentary failure this would produce ~30% muscle fatigue, or if exercise was performed to momentary failure using ~30% MVC then this would incur ~70% fatigue. However, whilst research has suggested that greater fatigue is evident following lighter- compared to heavier-load exercise (30% vs. 80% MVC, respectively; *Fisher, Farrow & Steele, 2017*), the relationship appears more complex than simply 100% minus the force requirements of the task relative to MVC. For example, in the study by Fisher, et al. exercise to momentary failure at 30% MVC produced a decrement in force production of 37.94%, which was greater than the 13.48% reduction after exercise using 80% MVC. In addition, when effort matched to task failure (confirmed by maximal values using rating of perceived exertion; RPE), lighter load exercise appears to produce a greater degree of discomfort compared to heavier-load exercise (*Fisher, Ironside & Steele, 2017*). However, these studies have considered the knee extensors and as such cannot be used to infer a relationship for the lumbar extensors. Further, they have also only considered male participants.

Whilst there is a growing body of literature comparing fatigability and muscular endurance between males and females, the data appears equivocal when considering different muscle groups as well as muscle action performed. For example, *Hunter et al. (2004)* reported similar fatigability of the elbow flexors when measured by time to task failure for a sustained submaximal isometric contraction (20% maximal voluntary contraction; MVC) for strength matched males and females. However, *Gentil et al. (2017)*, when comparing trained and untrained males and females, recently reported that females demonstrate a higher fatigue tolerance, when considering percentage reduction in isokinetic torque, than males of a similar training status when performing elbow flexion exercise. *Maughan et al. (1986)* has reported significantly greater time to failure for females

compared to males at lighter- but not heavier-loads (20%, but not 50% or 80% MVC for isometric knee extension; and 50%, 60% and 70%, but not 80% or 90% 1-repetition maximum for dynamic elbow flexion).

Considering the lumbar and trunk extensors specifically, Clark et al. (2003) reported similar muscular endurance performance for males and females for a dynamic trunk extension (TEX) task (women = 24.3 ± 3.4, men = 24.0 ± 2.8 repetitions at 50% MVC). However, when performing an isometric endurance test (time to task failure at 50% MVC), males demonstrated a briefer time to failure than females (105.4 ± 7.9 compared to 146.0 ± 10.9 s, respectively). The authors also reported a similar fatigue response (reported as median frequency slope using electromyography; EMG) between the lumbar extensors and the biceps femoris for females, whereas males showed a greater fatigue in the lumbar extensors than the biceps femoris. However, our own laboratory has recently reported no relationship between TEX endurance using the Biering-Sorensen test (as used by Clark et al.) and ILEX strength both for asymptomatic—and CLBP symptomatic—participants (Conway et al., 2016). As such there seems disparity between dynamic and isometric exercise and testing methods between males and females. Furthermore, the current body of research cannot be applied specifically to the fatigability of the lumbar extensor musculature in isolation.

Combined, improved understanding of the fatigability of the lumbar extensors, as well as the associated discomfort, in response to different loading strategies and the impact of sex would be useful in the prescription of sex specific exercise. At present recovery from fatiguing exercise from different loading strategies in men and women is not well explored, and the sex-based differences in fatigability may be of importance to determine optimal strategies for training and rehabilitation of the lumbar musculature (Hunter, 2016). As such the aims of the present study were to compare the fatigue responses of participants performing ILEX resistance exercise at heavier- and lighter-loads as well as compare these between male and female participants.

## METHODS

### Experimental approach to the problem

The relationship between, load, sex, and fatigue response was examined across 3 testing conditions using an ILEX machine (MedX, Ocala, FL, USA). A fatigue response test (FRT) was performed for the following conditions; heavy load (HL; 80% MVC), light load (LL; 50% MVC) and control (CON; no training). The conditions were separated by no less than 72 h, with the condition type (HL, LL, CON) being randomized to minimize order effect.

### Participants

Approval was granted from the University Health, Exercise, and Sport Science (HESS) ethics committee at the first authors' institution (ID No. 687). Previous work from our group using knee extension revealed a between-condition effect size (ES) of 1.86 (between HL and LL). Sample estimate was based upon this to determine participant numbers ($n$) calculated using G * Power (Faul et al., 2007; Faul et al., 2009). The calculations showed that a minimum of four participants were necessary to meet the required power of 0.8 at an

Table 1 Participant characteristics (mean ±SD).

| Group | Age (years) | Height (cm) | Weight (kg) |
|---|---|---|---|
| Males ($n = 9$) | $23.8 \pm 6.4$ | $176.7 \pm 6.2$ | $73.9 \pm 9.3$ |
| Females ($n = 8$) | $21.3 \pm 0.9$ | $170.5 \pm 6.1$ | $65.5 \pm 10.8$ |

alpha level of $p < 0.05$ for comparison between loading conditions. However, recruitment for each sex group (males and females) was increased to $n = 10$ to provide greater power for secondary analysis of between-sex comparisons. Recreationally active asymptomatic males ($n = 10$) and females ($n = 10$) with no previous training experience of the lumbar extensors were recruited (see Table 1 for characteristics). Prior to testing, participants completed an informed consent and a physical activity readiness questionnaire (PARQ). For the purpose of this study, the exclusion criteria included: individuals who suffered from a heart condition, history of CLBP, any contraindications to exercise identified on the participant PARQ, or any knee or hip conditions which prevented the use of the machine restraints.

## Testing procedures

Prior to testing, all participants attended a familiarization session where they were assessed for lumbar range of motion using a goniometer built in to the ILEX machine, and performed isometric testing using the ILEX machine every 12° beginning at 72° (full lumbar flexion) through 60°, 48°, 36°, 24°, 12°, and 0° (full lumbar extension) to allow them to experience the technique required and reduce any learning effect. Details of the full test protocol using the MedX ILEX machine and its restraint system have been documented elsewhere (*Graves et al., 1990*). In brief, participants were provided a specific dynamic warm-up of the lumbar extensors for ∼60 s using ∼27 kg for males and ∼20 kg for females. An MVC was then performed at seven different joint angles (as described above) by the participant gradually building force up to maximal effort over 3 s, and then relaxing over a further 3 s. Participants were provided with verbal encouragement to ensure maximal effort and were permitted ∼10 s recovery between testing angles. The ILEX machine and restraint system is shown in Fig. 1 and prevents rotation of the pelvis allowing training and testing of the lumbar extensors in isolation. The device has high test-retest reliability values of $r = 0.81–0.97$ in asymptomatic persons (*Graves et al., 1990*). An intra-class correlation coefficient (ICC) was conducted to assess pre-test MVC test-retest reliability at 72° between conditions (ICC = 0.931 (95% CI [0.845–0.972])) supporting the use of MVC for fatigue response testing in our laboratory.

## Fatigue response testing

Following the familiarisation session, participants were invited to return to the laboratory on three separate occasions (with not less than 72 h between) where they were assigned each testing condition (HL, LL, and CON) in a computer generated randomised order. The FRT required participants to complete the previously described testing procedure (pre-MVC) and then complete one of three conditions; a single set of ILEX exercise with a load equating to 80% of the maximum torque (HL), a single set of ILEX exercise with

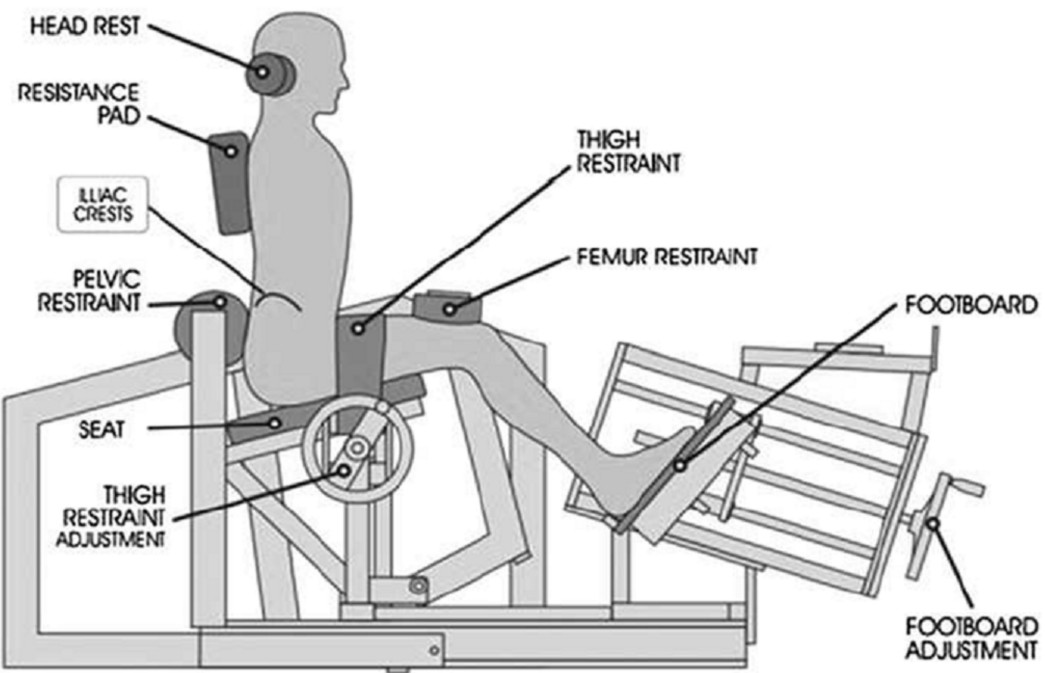

**Figure 1  Restraint system for the MedX Isolated lumbar extension machine.**

a load equating to 50% of the maximum torque (LL) or a control condition where the participants performed no exercise. Exercise was performed at 2 s concentric: 4 s eccentric (2:4) repetition duration to ensure standardisation with visual time feedback on a display in front of the participant. The ILEX machine also provides an audible sound at the completion of each phase of the repetition (e.g., at full flexion and full extension) to ensure the full range of motion is performed for each repetition. As fatigue incurred, the repetition duration generally increased but participants were encourage never to move faster than the predetermined repetition duration. The exercise was ceased when, despite their maximum effort, participants could not complete the concentric phase of a repetition (e.g., momentary failure; see *Steele et al. (2017a)* for a detailed description). Immediately (<10 s) following cessation of the dynamic exercise the participant repeated the above detailed isometric testing procedures (post-MVC) to assess the decrement in force production incurred as a result of the dynamic exercise. FRT testing using the ILEX machine has previously been reported elsewhere (*Edinborough, Fisher & Steele, 2016*).

As a result of pilot testing, a difference was identified for the time-under-load (TUL) between the HL and LL conditions. As such to ensure parity in time interval (~3 min) between the pre- and post-MVC testing following the HL, LL, and CON conditions, the dynamic exercise was delayed by 120 and 30 s for the HL and LL conditions, respectively. This rest interval was applied between the pre-MVC and exercise condition so that the post MVC could be performed immediately after the exercise condition. For the CON condition participants remained seated in the ILEX machine for 3 min before completing the post-MVC.

Immediately following each condition (but before the post-MVC), each participant was asked to report a rating of perceived exertion for effort (RPE-E) and discomfort (RPE-D) using 0–10 scales that permitted appropriate differentiation of the 2 perceptions (*Fisher, Farrow & Steele, 2017*; *Fisher, Ironside & Steele, 2017*; *Steele et al., 2017b*).

## Statistical analyses

Strength was considered as peak MVC which occurred at 72° for all participants. Fatigue was considered as the decrease in MVC as a result of the training condition in both absolute (post-MVC Nm—pre-MVC Nm) and relative units ([post-MVC Nm/pre-MVC Nm] × 100). The independent variable considered was the exercise condition (HL, LL, and CON), as well as sex (male and female), and the dependent variables included pre-MVC (the MVC prior to each condition), absolute and relative fatigue, time under load (TUL), and RPE-D. Analysis was not performed for RPE-E since all participants reported maximal effort (i.e., 10) at the cessation of dynamic exercise at both HL and LL.

Shapiro–Wilk test was conducted to examine whether data met assumptions of normality of distribution and Mauchly's test was used to examine assumptions of sphericity for repeated measures. For variables collected across all three conditions (pre-MVC, absolute, and relative fatigue), data met assumptions of normality and sphericity and thus a 3 × 2 repeated measures analysis of variance (ANOVA) was used to compare within participants across the within participant factor 'condition' and between participant factor 'sex'. A 2 × 2 repeated measures ANOVA was used in the case of TUL as it was only measured during HL and LL conditions. *Post hoc* pairwise comparisons with a Bonferroni correction were used for significant effects by 'condition', 'group', or 'condition × group'. RPE-D data did not meet assumptions of normality of distribution and thus a Wilcoxon test was conducted to examine differences between conditions. Mann–Whitney $U$ tests were used to compare dependent variables between sexes.

Further, within participant ESs using Cohen's $d$ (*Cohen, 1992*) were calculated for absolute and relative fatigue for each condition ($d = \mu_{change}/\sigma_{change}$), where an ES of 0.20–0.49 was considered as small, 0.50–0.79 as moderate and $\geq$0.80 as large. All statistical analyses were performed using IBM SPSS Statistics for Windows (version 23; IBM Corp., Portsmouth, Hampshire, UK) and $p < 0.05$ set as the limit for statistical significance.

## RESULTS

Three participants (two female and one male) were withdrawn from the study due to non-attendance of testing sessions leaving nine males and eight females for data analysis (participant characteristics are provided in Table 1).

Repeated measures ANOVA revealed no significant effects by 'condition' for pre-MVC ($F_{(2,30)} = 1.113$, $p = 0.342$), or interaction effects for 'condition × group' ($F_{(2,30)} = 1.609$, $p = 0.217$), though there was a significant effect by 'group' alone ($F_{(1,15)} = 10.663$, $p = 0.005$; estimated marginal means ± standard error, Males = 373.1 ± 20.7 vs Females = 274.3 ± 22.0). Pre-MVC by condition and sex are shown in Table 2.

Repeated measures ANOVA revealed a significant effect by 'condition' ($F_{(2,30)} = 44.252$, $p < 0.001$), 'group' ($F_{(1,15)} = 8.104$, $p = 0.012$), and 'condition × group' ($F_{(2,30)} = 5.248$,

**Table 2** Mean pre-MVC, absolute fatigue, and relative fatigue within groups and load conditions.

| Condition | Sex | Pre-MVC (Nm; mean ± SD) | Absolute change in MVC (Nm; mean ± SD) | Relative change in MVC (%; mean ± SD) |
|---|---|---|---|---|
| CON | Males | 355.2 ± 56.4 | 5.59 ± 20.7 | 1.3 ± 5.9 |
| | Females | 277.4 ± 45.4 | −2.55 ± 25.5 | −0.5 ± 9.7 |
| HL | Males | 369.6 ± 97.2 | −77.1 ± 41.9 | −21.3 ± 9.8 |
| | Females | 272.3 ± 58.5 | −32.6 ± 39.4 | −10.6 ± 14.0 |
| LL | Males | 394.4 ± 78.7 | −128.8 ± 41.4 | −33.3 ± 9.9 |
| | Females | 273.2 ± 57.0 | −70.7 ± 29.6 | −25.9 ± 8.4 |

**Notes.**

MVC, maximal voluntary contraction; CON, control condition; HL, heavier-load condition; LL, lighter-load condition; Nm, Newton metres.

$p = 0.011$) for absolute fatigue. *Post hoc* pairwise comparisons revealed significant between condition differences between CON and both HL ($p < 0.001$) and LL ($p < 0.001$), and between HL and LL ($p = 0.005$). ESs for absolute fatigue were small or negligible for CON (Males = 0.27, Females = −0.09), and large for both HL (Males = −1.84, Females = −0.83), and LL (Males = −3.11, Females = −2.39). Repeated measures ANOVA revealed a significant effect by 'condition' ($F_{(2,30)} = 63.560$, $p < 0.001$), but not for group ($F_{(1,15)} = 2.191$, $p = 0.160$), or 'condition × group' ($F_{(2,30)} = 2.938$, $p = 0.068$) for relative fatigue. *Post hoc* pairwise comparisons revealed significant between condition differences between CON and both HL ($p < 0.001$) and LL ($p < 0.001$), and between HL and LL ($p = 0.001$). ESs for relative fatigue were small or negligible for CON (Males = 0.22, Females = −0.05), and large for both HL (Males = −2.17, Females = −0.76), and LL (Males = −3.36, Females = −3.08). Both absolute fatigue and relative fatigue by condition and sex are shown in Table 2 and Fig. 2.

Repeated measures ANOVA revealed a significant effect by 'condition' ($F_{(2,15)} = 81.852$, $p < 0.001$), but not group ($F_{(1,15)} = 3.637$, $p = 0.076$), or 'condition × group' ($F_{(2,15)} = 3.017$, $p = 0.103$) for TUL. TUL by condition and sex is shown in Table 3.

Wilcoxon signed ranks test revealed a significant between condition difference for RPE-D ($Z = -3.568$, $p < 0.001$). However, Mann Whitney $U$ tests did not reveal any significant between sex differences for either HL ($Z = -0.264$, $p = 0.791$) or LL ($Z = -0.742$, $p = 0.458$). RPE-D values for condition and sex are shown in Table 4.

# DISCUSSION

The aims of the present study were to compare the fatigue responses of participants performing dynamic ILEX resistance exercise at heavier- and lighter-loads as well as compare these between male and female participants. The findings obtained from this study contribute to the current dearth in the literature of both sex comparative studies and fatigue following dynamic exercise performed to momentary failure.

## Acute fatigue

While similarly to our previous work (*Fisher, Farrow & Steele, 2017*) LL conditions resulted in both a higher absolute, as well as relative fatigue. Our analyses also revealed that males

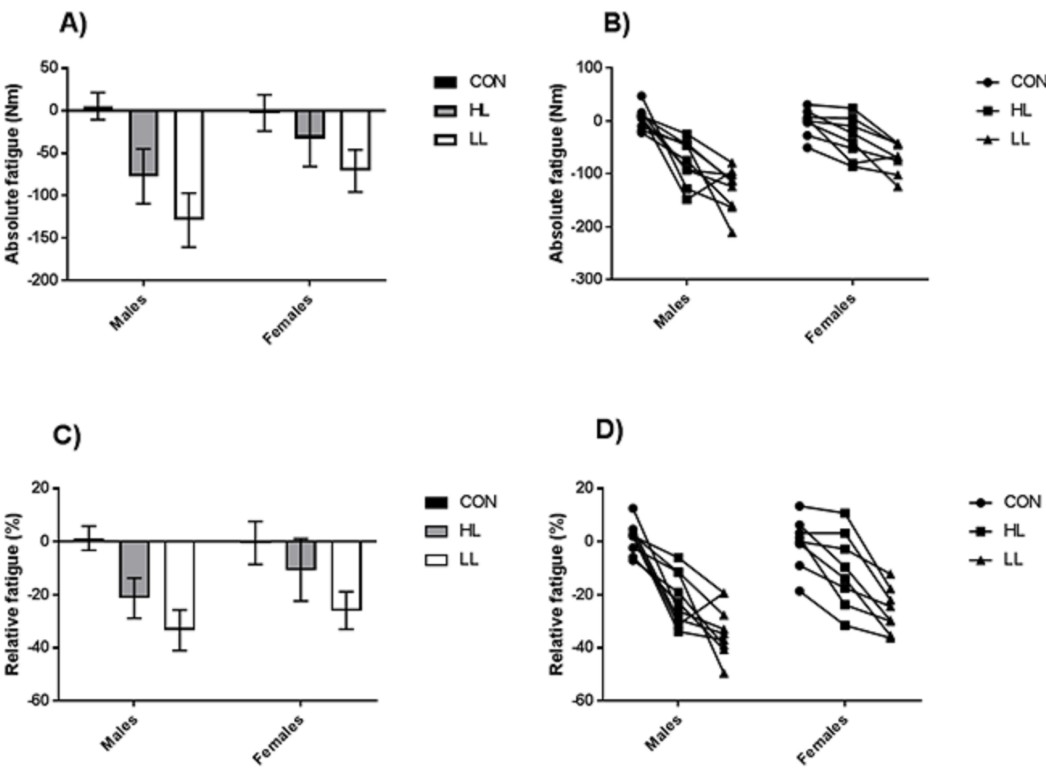

**Figure 2** Mean ±95% CIs (A) and individual responses (B) for absolute fatigue, and mean ±95% CIs (C) and individual responses (D) for relative fatigue, between conditions and sexes.

**Table 3  Mean (±SD) values for time-under-load between sex and condition.**

|         | Heavier-load (s) | Lighter-load (s) |
|---------|------------------|------------------|
| Males   | 57.7 ± 14.2      | 133.3 ± 29.7     |
| Females | 62.4 ± 17.6      | 174.0 ± 54.5     |

**Notes.**

MVC, maximal voluntary contraction; CON, control condition; HL, heavier-load condition; LL, lighter-load condition; Nm, Newton metres.

**Table 4  Mean (±SD) values for discomfort (RPE-D) between sex and condition.**

|         | Heavier-load   | Lighter-load   |
|---------|----------------|----------------|
| Males   | 6.33 ± 0.71    | 8.00 ± 0.71    |
| Females | 6.25 ± 0.71    | 8.25 ± 0.71    |

and females differ in their fatigability in the lumbar extensors. Significantly greater absolute fatigue was incurred by males performing exercise with the same relative load although, whilst still greater compared with females, differences in relative fatigue were not significant; possibly as a result of a type II error as previous research has supported sex differences in relative fatigue (*Gentil et al., 2017*). In the HL condition males showed a mean decrement in force production of 77.1 Nm (~21%) compared to 32.6 Nm (~11%) for females, whereas

following the LL condition males showed a mean decrease of 128.8 Nm (∼33%) compared to 70.7 Nm (∼26%) for females. Correspondingly, endurance capacity appears to differ between males and females, as time-under-load (time to task failure) was shorter for males than females in the LL condition (mean = 133.3 vs. 174.0 s, respectively), but not the HL condition (mean = 57.7 vs. 62.4 s, respectively). The present data suggests that, for the lumbar extensors, when exercising to momentary failure using a HL (80% MVC) males and females show similar time to task failure but show a large disparity in their absolute and relative fatigue following exercise. In contrast, at a LL (50% MVC) females showed greater endurance capacity by being able to continue exercising for ∼41 s (∼24%) longer than males, though showed a more similar decrement in force production (males = 33%, females = 26%).

The present findings agree with the body of literature suggesting women show a smaller decrement in maximal force production compared to males, following fatiguing exercise (*Hunter, 2016*). The greater TUL in females compared to males is also supported by studies showing sex-based discrepancies in endurance capacity during muscular contraction (*Maughan et al., 1986*; *Clark et al., 2003*; *Gentil et al., 2017*). However, where previous data suggested a similar number of repetitions prior to task failure between males and females for dynamic TEX exercise at 50%MVC (*Clark et al., 2003*), our data suggests that where the lumbar extensors are considered in isolation males show a shorter time to failure, and greater degree of fatigue.

Our data also adds to our understanding of fatigue at HL exercise. *Maughan et al. (1986)* showed that males and females show similar time to task failure at high-force or HL exercise (80% MVC for isometric knee extension; 80% and 90% 1-repetition maximum for dynamic elbow flexion). Whilst our data supports this regarding time to task failure (57.7 and 62.4 s for males and females, respectively) we have shown that, despite the similar duration of exercise, males incurred a greater absolute and relative degree of fatigue.

Differences between males and females for degree of fatigue and time to failure has been hypothesised to relate to differences in muscle fibre type (*Bajek et al., 2000*). Type I muscle fibres have a greater oxidative capacity which makes them markedly more fatigue resistant allowing contractions to be sustained over a longer period but of lower force output than type II muscle fibres. Indeed, our results revealed significant between sex differences in pre-MVC—supportive of greater strength in males compared to females. In addition, because the cross-bridge cycling is faster in type II muscle fibres, utilisation of adenosine triphosphate (ATP) occurs more rapidly than in type I muscle fibres (*Westerblad, Bruton & Katz, 2010*). Any disparity between HL and LL might be explained by the location of the onset of fatigue in relation to the $\alpha$-motor neuron. Research has suggested that LL activities likely incur elevated central motor output and as such incur higher levels of fatigue and greater reductions in maximal torque. With HL exercise fatigue appears to develop progressively and exercise cessation occurs as a result of centrally mediated factors (a decrease in the number and discharge of motor units). In contrast, fatigue as a result of LL activities appears to occur peripherally as a result of metabolic changes within the muscle preventing transmission of muscle action potentials (*Boyas & Guével, 2011*; *Gandevia, 2001*). Research has suggested women have either a greater relative percentage of type I

fibres compared to males (*Bajek et al., 2000*), or have a similar number but show a larger area of type I than type II muscle fibres (70–75% compared to 54–58%) when compared to males (*Thorstensson & Carlson, 1987*; *Mannion et al., 1997*). The lumbar extensors are postural muscles and, as such, it would be expected that type I fibres predominate in this group due to a need for sustained or repeated lower force actions. However, this disparity in relative percentage and/or size of type I fibres might explain the greater fatigue resistance in females compared to males observed in the present study, both in strength, the degree of fatigue incurred at both HL and LL, and the time to task failure in LL. However, we should acknowledge that the present study was not intended to assess muscle fibre type disparity or causal factors of fatigue, and as such, we only speculate on potential underpinning mechanisms.

From an applied perspective it is important to recognise the differences in performance and fatigue decrement between males and females. Our data suggests that females do not incur the same degree of fatigue as males following similar exercise protocols, and indeed that females might be able to sustain longer exercise duration at the same relative loads when using low, but not high loads. As such females should manipulate training variables accordingly when training with high loads, performing either greater repetitions at the same relative load, or using heavier relative loads than males. Furthermore, if females do not show the same performance decrement then it might be feasible that females can train at a higher frequency than males to optimise adaptation. However, it should be noted that in the present paper we have only examined the acute fatigue response. Previous research using a larger training volume (four sets of 10-repetitions @10-repetition maximum, followed by four sets of 10 repetitions @ 80% of 10-repetition maximum) resulted in a similar reduction in peak torque of the elbow flexors between males and females. However, data suggested a longer time to return to baseline torque for females compared to males (*Flores et al., 2011*). Whilst the relevance to the present study is limited since there is considerable disparity in exercise volume between the studies, future research might consider recovery from fatiguing exercise between males and females using different exercise volumes and across different muscle groups.

## Effort and discomfort

No analyses were performed for RPE-E since all participants reported maximal values as a result of training to momentary failure (*Steele et al., 2017a*). This is supportive of previous research which has shown that effort and discomfort can be distinguished independently (*Fisher, Ironside & Steele, 2017*; *Steele et al., 2017b*). Performing repetitions to momentary failure in the LL condition induced greater discomfort compared to the HL condition. This is also in accordance with research considering the knee extensors where effort and discomfort have been assessed independently (*Fisher, Ironside & Steele, 2017*). Our analyses also showed no sex-based differences for RPE-D ($p > 0.05$). Previous research has reported varying but submaximal values for effort when assessed alone (*Shimano et al., 2006*). However, it has previously been discussed that where exercise is continued to the point of momentary failure effort should be maximal irrespective of load, time-under-load, repetitions completed prior to task failure, etc., and that variation in perceptual responses

are likely a product of discomfort (*Smirnaul, 2012*; *Steele et al., 2017b*). In this sense it is important to be able to anchor effort using the RPE-E and independently assess discomfort (RPE-D), and vice-versa, as we have done so in the present article.

As stated, when performing LL resistance exercise momentary failure appears to occur as a result of peripheral fatigue (*Boyas & Guével, 2011*; *Gandevia, 2001*). This prolonged ATP production incurs greater metabolite accumulation in the form of increased inorganic phosphate ($P_i$) accumulation of hydrogen ions ($H^+$) and thus decreased intramuscular pH (*Schott, McCully & Rutherford, 1995*; *MacDougall et al., 1999*; *Takada et al., 2012*). Perceptions of discomfort appear to be linked to afferent feedback and thus a reason why increased metabolic stress may promote greater discomfort during LL resistance exercise (*Marcora, 2009*). Contrastingly, perceptions of effort are likely related to central motor output (*De Morree, Klein & Marcora, 2012*) which may explain the similarly maximal RPE-E that participants gave for both LL and HL when performed to momentary failure. Indeed, it has been argued that both LL and HL likely result in similar numbers of motor units being recruited, albeit with differing recruitment strategies (*Fisher, Steele & Smith, 2017*), and thus likely a similar central motor command being required upon achieving momentary failure.

However, although higher discomfort levels were reported following the LL condition, we can again only speculate upon the potential influences of central or peripheral response. The results from our study may however be useful in a practical sense. Since greater discomfort was incurred in alignment with greater decrements in MVC as a result of the LL condition, and knowing that similar adaptations in muscular strength can occur using both heavier- and lighter-loads of resistance (*Assunção et al., 2016*; *Schoenfeld et al., 2017*; *Fisher, Ironside & Steele, 2017*; *Fisher, Steele & Smith, 2017*), individuals may be more inclined to exercise using a heavier load since it appears to incur a lower level of discomfort.

## Limitations

The present study adds to the dearth of literature considering female participants and comparing sex-based fatigue. However, it is important to recognise any potential limitations within the present study that might be considered for future research. Whilst a power analysis confirmed adequate sample size for between condition comparisons, only 17 participants completed the study and as such between sex comparisons may have been underpowered (possibly explaining the lack of significant interaction effects for relative fatigue; indeed, *post hoc* observed $\beta$ for this was 0.318). Future work might consider the typical effect size differences between sexes in determining sample estimates, and also might consider what constitutes a clinically meaningful difference. Further, our results may be limited in application to persons with the characteristics that conformed to our inclusion criteria. Whilst we have discussed the efficacy of strengthening the lumbar extensors to reduce CLBP there were no participants involved in the study with the condition. As such it would be unreasonable to assume these findings would be equated with this particular population, therefore it may be beneficial to replicate this study using both CLBP symptomatic and asymptomatic individuals.

## CONCLUSION

The present study advances the limited research in acute responses of fatiguing exercise to differing loading strategies, notably comparing males and females, and assessing the lumbar extensors in isolation. The findings revealed load dependent disparities, in addition to sex based differences, in fatigue response as a result of ILEX exercise performed to momentary failure. Since females do not incur the same degree of fatigue as males following similar exercise protocols, females might manipulate training variables accordingly. Albeit speculative, it might be more efficacious for females to perform a larger number of repetitions for the same relative load, or use a greater relative training load compared to males to increase volume-load or maximise strength-specific adaptations. Furthermore, if a lesser decrement in performance is incurred from a similar training volume then females might consider either (a) increased training volume, or (b) greater training frequency, both of which might result in greater calorific expenditure which might favourably optimise body composition adaptations. However, the latter is recommended tentatively as the differences between sexes in rate of recovery to baseline after RT is not clear.

Previous literature has reported similar increases in muscular strength as a result of HL and LL resistance exercise (*Assunção et al., 2016*; *Schoenfeld et al., 2017*; *Fisher, Ironside & Steele, 2017*; *Fisher, Steele & Smith, 2017*) and as such our consideration of discomfort relating to load is important. Indeed, previous research has shown that low-volume (1 day/week) ILEX resistance exercise can strengthen the lumbar extensors (*Fisher, Bruce-Low & Smith, 2013*) and as a result, reduce CLBP (*Bruce-Low et al., 2012*; *Steele, Bruce-Low & Smith, 2015*). Lighter load exercise is often prescribed in rehabilitation settings perhaps owing to clinician's perceptions that it may be less likely to result in pain. Indeed, clinicians in general seem reluctant to recommend high intensity of effort exercise either (*Munneke et al., 2004*). However, in addition to exercise related discomfort, group III/IV muscle afferents are involved in nociception and thus may be involved in perception of pain (*McCord & Kaufman, 2010*). It seems prudent for clinicians to know firstly that, effort and load are not synonymous and, given correct instruction, people can differentiate effort and discomfort; and secondly that lighter load exercise might be contraindicated due to the increased fatigue and discomfort associated with it. Discomfort associated with LL exercise might be misinterpreted by a patient as pain relating to their injury. However, tools exist allowing patients to differentiate the qualities of different pain experiences (*Melzack & Katz, 2001*) and so it may be the case that with correct instruction patients can differentiate exercise related perceptions of discomfort from perceptions of pain associated with their injury. Future research might consider the use of HL and LL resistance exercise in the lumbar extensors in rehabilitation environments to assess the efficacy of the load-based training prescription with careful consideration to the perceptual elements of effort, exercise related discomfort, and injury related pain.

### Funding
The authors received no funding for this work.

### Competing Interests
The authors declare there are no competing interests.

### Author Contributions
- Charlotte Stuart conceived and designed the experiments, performed the experiments, analyzed the data, prepared figures and/or tables, authored or reviewed drafts of the paper, approved the final draft.
- James Steele performed the experiments, analyzed the data, contributed reagents/materials/analysis tools, prepared figures and/or tables, authored or reviewed drafts of the paper, approved the final draft.
- Paulo Gentil and Jürgen Giessing prepared figures and/or tables, authored or reviewed drafts of the paper, approved the final draft.
- James P. Fisher conceived and designed the experiments, performed the experiments, analyzed the data, contributed reagents/materials/analysis tools, prepared figures and/or tables, authored or reviewed drafts of the paper, approved the final draft.

### Human Ethics
The following information was supplied relating to ethical approvals (i.e., approving body and any reference numbers):

The Health, Exercise and Sport Science committee at Southampton Solent University granted Ethical approval to carry out the study within its facilities (Ethical Application ID No. 687).

### Data Availability
The raw data are provided in a Supplemental File.

### Supplemental Information
Supplemental information for this article can be found online at http://dx.doi.org/10.7717/peerj.4523#supplemental-information.

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
