# Peer review of "Fatigue and perceptual responses of heavier- and lighter-load isolated lumbar extension resistance exercise in males and females"

_PeerJ, doi:10.7717/peerj.4523_

## Round 0.1 · original submission · Minor Revisions

Thank you for your submission of this well written manuscript, based on a well conducted study. Please see the comments from the reviewers. Upon resubmission, please address each of the reviewers' comments with your changes or rebuttal. I look forward to receiving the next draft.

Scotty

·

Basic reporting

The authors have provided a succinct, clear report of their study with well described methodology and rationale. Thank you for providing the raw data. The group codes in the supplementary file currently use numerical values (1 vs 2), that could be unmasked to what I assume is male and female based on the values.

Experimental design

No comment, adequately designed and described methodology.

Validity of the findings

Line 343-345: Participant flow is adequately documented in the study, and that the target sample size was not attained. Consequently, the between-sex comparisons may be under powered. This could be substantiated by providing a post hoc power calculation and discussion as to what degree of difference one should consider meaningful between the sexes. Is there a clinically important difference for fatigue values, or a difference that at least exceeds the measurement error such that we would need to consider the power of the comparison?

Additional comments

-Lines 355-358: The authors conclude that based on the differential fatigue response between the sexes, that training program modifications may be required based on sex. It is not necessarily clear what outcomes would be expected to be different between males and females in this instance. Many studies have demonstrated comparable hypertrophic and improved (but differential) strength responses with varying training loads taken to the point of failure, would there be other relevant adaptations that should be considered in the context of this data or muscle group? Discussion on this may be somewhat speculative in nature and beyond the scope of the article but would enhance the existing statement.

-Line 27/28: Inconsistent spacing between numerical symbols and numbers, this is also evident in figure legends and tables (PDF rendering issue?)

-Line 186: Unnecessary footnote, could be included in the paragraph

-Line 369/ Line 45: Fatigue occurred following both high and low load exercise, but only in the low-load condition is fatigue referred to as "debilitating". What differentiates the use of the term "fatigue" and "debilitating fatigue". Stating "increased fatigue" may be more appropriate.

-Line 367-70: Is there a common misconception in rehabilitation settings that low-training is required to strengthen the musculature? This may have less to do with a relationship to desired training adaptations and relate more to clinician perceptions of loading tolerance/pain. The data on fatigue and discomfort is useful clinically, however, is there any connection between RPE-D and pain (i.e. NPRS)? Is the perception of clinicians that the use of low-loads reduces the risk of increasing pain relative to high loads, irrespective of discomfort?

-Line 371: "....in other rehabilitation environments". It is not clear what rehabilitation environment this was completed in. Were they asymptomatic, healthy controls? If so, should read "....rehabilitation environments"

Reviewer 2 ·

Basic reporting

Reporting is overall very well done.
Abstract: final sentence - lighter load is in fact used in rehabilitative settings to increase 'endurance' of the lumbar extensors as an extension from Biering Sorensen's original finding of decreased endurance and its association with back pain. However, as load is required for strength the statement is flawed. Lighter load may increase endurance, but not strength. Please revise the sentence.

Some disparity in the wording of the aims in the final sentence of the introduction and the first sentence of the discussion. Restating may be easier for the reader.
Table 1 - please change Kg to kg (SI units)
Line 156 - Comma after incurred
Line 158 - Comma after effort
Line 275 - A reference is needed in here.
Line 442 - Capitalize Journal

Experimental design

Line 130 - How was lumbar ROM assessed? Seeing there are numerous ways of assessing this, some explanation for reproduction may be warranted.

Validity of the findings

Not a criticism as the authors have reported on both central and peripheral influences to fatigue, some of Professor Samuel Marcora's newer work on the central influence of fatigue (if its published yet) could be added. This could be an nice addition to the discussion around Line 321.

Additional comments

Congratulations on a well written paper. Some minor modifications are suggested.

---

## Round 0.2 · accepted · Accept

Thank you for your attention to the reviewer's comments and for addressing their suggestions. I agree with your rebuttal of reviewer 2's comments on the nature of low load/endurance vs strength for rehabilitation purposes. I believe that research such as yours will go a long way to help rehab practitioners improve their knowledge and skill in obtaining physiological effects with therapeutic exercise.

Congratulations!
Scotty